# Extracellular Vesicle-Based Therapeutics for Heart Repair

**DOI:** 10.3390/nano11030570

**Published:** 2021-02-25

**Authors:** Laura Saludas, Cláudia C. Oliveira, Carmen Roncal, Adrián Ruiz-Villalba, Felipe Prósper, Elisa Garbayo, María J. Blanco-Prieto

**Affiliations:** 1Department of Pharmaceutical Technology and Chemistry, Faculty of Pharmacy and Nutrition, University of Navarra, 31008 Pamplona, Spain; lsaludas@alumni.unav.es; 2Instituto de Investigación Sanitaria de Navarra (IdiSNA), 31008 Pamplona, Spain; croncalm@unav.es (C.R.); fprosper@unav.es (F.P.); 3Department of Animal Biology, Institute of Biomedicine of Málaga (IBIMA), Faculty of Science, University of Málaga, 29010 Málaga, Spain; claudia.ol.556@gmail.com (C.C.O.); adrian.ruizvillalba@gmail.com (A.R.-V.); 4Andalusian Centre for Nanomedicine and Biotechnology (BIONAND), 29590 Málaga, Spain; 5Laboratory of Atherothrombosis, Program of Cardiovascular Diseases, CIMA, University of Navarra, 31008 Pamplona, Spain; 6Centro de Investigación Biomédica en Red (CIBERCV), Carlos III Institute of Health, 28029 Madrid, Spain; 7Program of Regenerative Medicine, CIMA, University of Navarra, 31008 Pamplona, Spain; 8Cell Therapy Area and Haematology Department, Clínica Universidad de Navarra, 31008 Pamplona, Spain; 9Centro de Investigación Biomédica en Red (CIBERONC), Carlos III Institute of Health, 28029 Madrid, Spain

**Keywords:** cardiovascular diseases, myocardial infarction, cardiac repair, extracellular vesicles, exosomes, drug delivery, cargo loading, targeting

## Abstract

Extracellular vesicles (EVs) are constituted by a group of heterogeneous membrane vesicles secreted by most cell types that play a crucial role in cell–cell communication. In recent years, EVs have been postulated as a relevant novel therapeutic option for cardiovascular diseases, including myocardial infarction (MI), partially outperforming cell therapy. EVs may present several desirable features, such as no tumorigenicity, low immunogenic potential, high stability, and fine cardiac reparative efficacy. Furthermore, the natural origin of EVs makes them exceptional vehicles for drug delivery. EVs may overcome many of the limitations associated with current drug delivery systems (DDS), as they can travel long distances in body fluids, cross biological barriers, and deliver their cargo to recipient cells, among others. Here, we provide an overview of the most recent discoveries regarding the therapeutic potential of EVs for addressing cardiac damage after MI. In addition, we review the use of bioengineered EVs for targeted cardiac delivery and present some recent advances for exploiting EVs as DDS. Finally, we also discuss some of the most crucial aspects that should be addressed before a widespread translation to the clinical arena.

## 1. Introduction

Cardiovascular diseases (CVDs) include a wide diversity of pathologies of the heart and blood vessels such as coronary artery disease, rheumatic heart disease, cerebrovascular disease, ischemic heart disease, and heart failure [1]. Among them, myocardial infarction (MI) and stroke remain the highest cause of mortality globally [2]. Furthermore, they are included in the top-ten ranked diseases associated with age, sex, and territories in the last study of the Global Burden of Disease, which included 369 diseases and injuries, and 204 countries and territories [3]. Despite major advances in pharmacology and device surgery, CVDs remain a significant public health challenge. Their prognosis is poor, especially for MI. The majority of patients who survive their first MI have a high chance of recurrent MI or other complications [4]. The origin of these complications derives from the absence of a regenerative capacity in adult mammal hearts. After damage, a fibrotic scar replaces the native tissue. In the early stages after injury, this healing fibrotic scar is essential to cover the lost tissue. However, this scar becomes disabling and rigid with time. Resident cardiac cells are not able to regenerate the heart tissue and efficiently restore cardiac function after heart failure. Consequently, cardiomyocytes located in the border zone suffer hypertrophy because of a pressure overload, giving rise to cardiac function reduction and chronic cardiac damage. The most decisive treatment currently available is cardiac transplant, which is subject to considerable main limitations as it depends on the availability of donors.

In the past 20 years, regenerative cardiovascular medicine has developed several therapeutic interventions to address this problem, including the application of cell-based therapies [5]. Ideally, transplanted cells could engraft, proliferate, and differentiate into healthy new tissue [6]. Two main cellular sources have been used for this purpose: adult multipotent and pluripotent stem cells. On one hand, different adult multipotent stem cells have been transplanted into the infarcted heart: skeletal myoblasts, bone marrow mononuclear cells, CD34^POS^ circulating endothelial progenitors, mesenchymal stromal cells (MSCs) and their derivatives (cardiopoietic cells and mesenchymal precursor cells), and cardiosphere-derived cells (CDCs) [7,8]. All these cells showed some limitations in their differentiation capacity in vivo, mainly because of a reduced cell survival rate and engraftment in the fibrotic scar. In contrast, pluripotent stem cells, such as embryonic stem cells (ESCs) or induced pluripotent stem cells (iPS) have successfully given rise to different cardiac cell types after transplantation, although they found some immunogenic rejection [9]. However, both cellular sources have significantly contributed with the secretion of several paracrine factors to cardiac repair [8]. Between them, extracellular vesicles (EVs) have been recently reported as a promising approach for cardiac cell therapy [8,9].

The term EVs, as coined by the International Society of Extracellular Vesicles (ISEV), includes all extracellular membrane-enclosed vesicles [10]. Structurally, EVs are nanoscale cell-derived lipid vesicles that are secreted by virtually all known cell types under normal or pathological conditions, including cardiac cells [5,11,12]. EVs contain DNA, different RNA species, lipids, growth factors, metabolites, and proteins. Interestingly, a large volume of evidence indicates that the nature and the relative content of this cargo vary according to the producing cell type and its physiological state [13,14]. EVs play a role in intercellular communication inducing signaling via receptor–ligand interaction or through internalization by the recipient cell, delivering their content into the cytosol and modifying the cells’ physiological state [15].

In the context of cell therapy for cardiac repair, EVs show potential advantages over cells, such as the absence of tumorigenicity, lower immunogenic potential, product stability, non-limiting dosage by microvascular plugging or loss of transplanted cell viability, and the existence of multiple approaches to enhance efficacy, including genetic engineering of the parent cells [7,12]. For this reason, EVs purified from defined cell types have been investigated as novel therapeutic options for various cardiac diseases including ischemic heart disease and heart failure, as well as for pathogen vaccination, immune-modulatory and regenerative therapies, and drug delivery [16]. 

In this regard, bioengineering has been proposed for improving EVs’ potential for CVD treatment. Direct encapsulation of functional cargos or drug molecules has been demonstrated as an effective therapeutic approach to protect cells from drug toxicity and endogenous material from degradation [17,18]. Besides, the surface of EVs can be modified to improve their capacity regarding target presentation to specific cardiac cell types [5]. Altogether, and in comparison with conventional drug delivery systems (DDS), (pre)designed EV-based therapies could improve cellular and tissue distribution of the bioactive molecules and their efficacy, biocompatibility, immunogenicity, and also reduce their toxicity [19,20].

In this review, we discuss the potential of EV-based therapies for heart repair. First, we describe the biology of EVs, detailing their biogenesis, cargo, mechanisms of action, and the most current classification following the criteria established by the ISEV. Next, we present the therapeutic potential of native EVs for cardiovascular applications based on their cellular origin. Finally, we summarize the most relevant studies published in the last five years related to the potential loading mechanisms, surface modification/engineering to target specific cardiac cells, and the use of EVs as new DDS for MI, discussing the challenges and the perspectives of this nascent field.

## 2. Extracellular Vesicles

### 2.1. Classification

EVs comprise submicron particles heterogeneous in size, delimited by a lipid bilayer that cannot replicate. Traditionally, they have been classified according to their size and biogenesis, distinguishing: small particles or exosomes of endosomal origin with diameters ranging from 30 to 150 nm; ectosomes or microvesicles directly shed from the plasma membrane and polydisperse in size (100–1000 nm); and apoptotic bodies generated as a consequence of programmed cell death (1000–5000 nm) [10,21]. However, in recent years it has become apparent that the picture is more complex than expected. Assigning an EV to a particular biogenesis pathway still remains extraordinarily difficult given the overlap in size-distribution and protein-expression patterns among different EV types, especially when referring to exosomes and microvesicles, challenging the attempts to define a more precise nomenclature for EV classes [22]. Consequently, the latest recommendations of the ISEV encourage authors to define EV subtypes considering their physical characteristics—attending to: (a) size (small, medium/large) or density (low, medium, high) according to a defined range, (b) biochemical composition relaying on specific markers (e.g., CD63, CD81, annexin V, etc.), (c) isolation conditions (e.g., hypoxia, serum conditioning), and/or d) cellular origin (platelet, endothelial, cardiomyocytes, etc.)—rather than by the use of the traditional terms exosomes or microvesicles [10]. A summary of EVs’ physical and biochemical properties as well as parental cell conditions is provided in Table 1. In this review, the term EVs will refer to both exosomes and microvesicles. 

### 2.2. Biology

EVs are released into the extracellular space by most cell types and can be found in a wide range of body fluids as reservoirs of lipids, proteins, nucleic acids, and carbohydrates of their parental cells. Current knowledge supports the view that each cell type tunes EVs’ biogenesis, depending on its activation status. Moreover, their cargo is particular to the stimulus or biological condition triggering their formation and release, suggesting the existence of intracellular selective cargo-sorting mechanisms. Subsequently, EVs’ composition will directly affect their fate and function [21,27,28]. There is a consensus about two major EV biogenesis pathways, giving rise to the most widely studied subpopulations, exosomes and microvesicles. Exosomes, generated within the endosomal system, are intraluminal vesicles formed by the inward budding of the endosomal membrane during maturation of multivesicular endosomes (MVEs) and are released to the extracellular space by fusion of MVEs with the cell membrane [29]. As for microvesicles, they originate by outward budding of the plasma membrane [30]. Microvesicle biogenesis determines the expression of surface-specific antigens from the cell origin, as well as the externalization of phosphatidylserine on the outer membrane leaflet [31], although the latter is not a prerequisite in all microvesicles [32]. Despite the aforementioned differences in biogenesis, the overlap in size, density, or composition, together with the lack of appropriate technology, hampers the possibility of distinguishing EV subpopulations once released to the extracellular medium, favoring the use of the generic term EVs, instead of a more specific nomenclature [10].

### 2.3. Mechanism of Action

The lipid bilayer protects EVs’ content from degradation by nucleases and proteinases present in biofluids, enabling the transfer of proteins, lipids, or nucleic acids from parental cells to recipient cells [33]. As such, EVs contribute to normal homeostasis, but also to the progression of several pathologies, including CVDs [13,34]. The mechanisms by which EVs mediate intercellular communication are not completely understood but are supposed to involve specific interactions between proteins or lipids enriched at the EVs surface (e.g., tetraspanins, integrins, lectins, phosphatidylserine) and receptors at the plasma membrane of recipient cells (e.g., intercellular adhesion molecules (ICAMs), annexin V, galectin 5) [21,35,36,37,38]. After docking at the cell membrane, EVs can remain at the binding site eliciting functional responses in recipient cells by activating downstream molecular pathways, or by direct interaction with extracellular matrix components [39] (Figure 1). They can also be internalized by endocytosis or by fusion with the plasma membrane undergoing different fates. For instance, endocytosed EVs can reach the MVEs and be targeted for degradation by lysosomes, they can escape digestion by back fusion with the MVEs’ membrane, or they can be re-secreted to the extracellular space via the early endocytic recycling pathway [39,40,41,42,43]. Either by direct fusion with the plasma membrane or after escaping lysosomal degradation, EVs can release their content into the cytoplasm of recipient cells and regulate cellular processes [44] (Figure 1).

### 2.4. Separation and Characterization of Extracellular Vesicles

EVs can be separated from the cell culture medium and most body fluids (liquid biopsy), blood being the most frequently studied source of EVs. Before EV separation, some preanalytical parameters should be considered [45,46], such as the use of serum-free media or EV-depleted serum for EV separation from conditioned medium [10,47]. Moreover, differences in the physicochemical and biochemical properties of the selected separation methods can impact the enriched EV subpopulations [48,49,50,51,52] and do not enable an absolute purification of EVs from other contaminants [53].

Ultracentrifugation (UC) is the most commonly used EV separation and enrichment technique based on particle density, involving multiple centrifugation and ultracentrifugation steps [48]. Speeds of 10,000–20,000 g enable the separation of medium/large vesicles, while small-sized vesicles are recovered at higher speeds (100,000 g).

Size exclusion techniques include ultrafiltration and chromatography. Ultrafiltration is usually based on cellulose filters defined by molecular mass and size exclusion range [49], while size exclusion chromatography (SEC), separates fractions by elution with phosphate buffered saline (PBS) and has been proven to be reliable and scalable for various applications [54]. 

Immune affinity isolation is based on the immunolabeling of proteins on the surface of EVs, enabling the separation of specific particle subpopulations from other EV classes, contaminant protein aggregates or lipoproteins. Usually, specific antibodies are conjugated to magnetic beads and EVs are separated using magnets [55,56].

A range of commercial kits are also available, some based on polymer precipitation- methods [49,50,51] and others on non-precipitation alternatives, for instance, those selective for phosphatidylserine positive vesicles [47]. Alternative or complementary techniques to classical procedures are also emerging, including microfluidics, asymmetric flow field-flow fractionation, or high-resolution flow cytometry [53]. 

After separation, EVs’ purity should be tested by the use of multiple complementary methods: (i) western blotting to analyze EVs markers (e.g., CD63, Alix, etc.) and co-isolated contaminants [10]; (ii) nanoparticle tracking analysis (NTA), which can determine particle size and concentration [57]; (iii) conventional transmission electron microscopy (TEM) and the more strongly recommended cryo-TEM [58], or (iv) nanoflow cytometry, that enables the determination of cell surface antigens, the quantification of EV subpopulations based on parental cell markers [59,60], and the lipid nature of the studied particles with cell-permeant, non-fluorescent pro-dyes [61]. Also, current advances in EV-adapted proteomic, lipidomic, and genomic technologies will greatly help to delimit the molecular signature of the EV subpopulations under research.

## 3. Potential Applications of Extracellular Vesicles as Therapeutic Agents in Myocardial Infarction

Driven by the drawbacks associated with cell transplantation as well as the key role of stem cell paracrine secretion in cardiac repair, EVs have emerged recently as a next-generation cell-free regenerative therapy. Several studies have been performed in the last five years, aiming to test EVs’ potential as cell substitutes in the cardiac regenerative field, with significant preclinical success (Figure 2). All these studies collect different cell sources, isolation techniques, therapeutic doses, or administration routes, reflecting the heterogeneity and immature nature of the field. Here, we group the most relevant findings from these studies as well as a brief compendium of the EV-associated molecules involved in heart repair, based on the parental cell type. A summary of these can be found in Table 2.

### 3.1. Mesenchymal Stromal Cell-Derived Extracellular Vesicles

Undoubtedly, MSCs have been in the spotlight of cell-based therapies for recovering the ischemic myocardium after MI. MSCs are well-described stem cells that can be obtained from several body tissues and exert multiple regenerative effects due to their complex and enriched paracrine secretion [92,93]. Therefore, most of the recently published articles in this field exploit the therapeutic benefits associated with MSC-derived EVs (MSCs-EVs).

EVs’ cargo reflects in part the content of parental cells and thus it can exert comparable beneficial responses than their cellular counterparts in target cells and tissues (Figure 2). Accordingly, bone marrow MSC (BM-MSC)-derived EVs induce a similar cardioprotective effect to their parent BM-MSCs, disclosing their pivotal role in paracrine signaling and regenerative mechanisms [62]. Among their multiple properties, it has been recently described that BM-MSC-EVs possess immunomodulatory properties, favoring an anti-inflammatory macrophage phenotype polarization in infarcted mice via micro-RNA (miRNA)-182 delivery and inhibition of toll-like receptor 4 (TLR4) in recipient cells, which has a key impact on cardiac repair. Interestingly, macrophage depletion eradicated the beneficial effects, postulating inflammation modulation as the pivotal mechanism of cardiac protection [63]. In connection with that, Xu et al. found that the potential of these vesicles to reduce inflammation and induce a regenerative macrophage polarization was increased when EVs were obtained from pro-inflammatory BM-MSCs, by pre-incubation of cells with low concentrations of lipopolysaccharide, compared to EVs from non-altered BM-MSCs. However, cardiac functional studies are lacking, making it impossible to confirm a superior reparative effect [64].

The principal challenge of cardiac cell therapy is overcoming the low cell survival and retention in the heart due to a detrimental inflammatory and oxidative microenvironment after MI. Relying on the EVs’ immunomodulatory properties and aiming to create a more favorable milieu for subsequent stem cell transplantation, BM-MSC-EVs were locally injected into the rat heart 30 min after ischemia [65]. EV injection led to decreased inflammation, enhancing the retention and survival of BM-MSCs intravenously infused three days after infarction. Importantly, EVs also promoted an increase in stromal cell-derived factor 1 (SDF-1) concentration in the heart, which, along with the overexpression of its receptor CXC chemokine receptor 4 (CXCR4) in BM-MSCs by pre-treatment with atorvastatin, increased cell homing to the myocardium [65]. The interaction between SDF-1 and CXCR4 has been investigated and exploited in other studies as a relevant axis mediating cell recruitment in ischemic cardiomyopathies [94,95]. The administration of EVs secreted from BM-MSCs pre-treated with atorvastatin also induced better heart recovery than control BM-MSC-EVs in other studies [66]. However, in this case, the enhanced repair was attributed in part to a 13-fold increase in lncRNA H19 concentration in secreted EVs, a mediator involved in angiogenesis [66].

Another well-described reparative effect attributed to MSC-EVs is the formation of new blood vessels. Wang et al. confirmed that the intravenous (IV) delivery of BM-MSC-EVs to infarcted mice restored cardiac function due to an increased vascular density mediated by the delivery of miRNA-210 [67]. In agreement with this, Zhu et al. found in a more recent study that this EV-derived miRNA-210 was fundamental for cardiac repair. However, it was not involved only in the stimulation of angiogenesis but also in the reduction of infarct size and cardiomyocyte apoptosis, activation of resident progenitor cells and ultimately in the improvement of cardiac function [68].

MSC-EVs may also contribute to enhanced cardiac performance by other mechanisms. For example, Liu et al. achieved a decrease in myocardial ischemia/reperfusion (I/R) injury by administering BM-MSCs exosomes in a rat MI model [69]. Reperfusion is a double-edged sword that enables blood restoration to ischemic areas but also induces a large local oxidative stress that causes cardiomyocyte death and infarct expansion. In this case, EV injection before reperfusion induced an increase in cardiomyocyte autophagy via AMPK and Akt pathways, resulting in reduced cell apoptosis and infarct size [69]. In consonance, in another study, BM-MSC-EVs attenuated I/R injury in mice and provided cardioprotection by miRNA-21a-5p delivery, which was involved in decreasing infarct size [70].

Regarding cell culture conditions, Zhu et al. observed that the reparative effect of EVs in mice was not the same when vesicles were obtained from BM-MSCs cultured in normoxia or hypoxia, with the latter exerting a greater beneficial effect. Importantly, these authors went a step further and were able to attribute the differences observed to the enrichment of miRNA-125b in hypoxic vesicles [71]. A similar phenomenon was also described in other studies, where the superior therapeutic effect of hypoxic BM-MSC-EVs was attributed to an increased EV enrichment with different miRNAs, e.g., miRNA-210 [68] or miRNA-24 [72], accounting for the possible diverse miRNA players.

Beyond bone marrow, other body tissues have been also investigated as potential sources for MSCs and their associated EVs in cardiac repair. Adipose tissue represents one of the most appealing sources for MSCs, known as adipose tissue-derived mesenchymal stem cells (ADSCs), due to its abundance and easy collection by non-invasive techniques. Although less widely explored, ADSC-derived EVs (ADSCs-EVs) have also shown encouraging results, inducing multiple reparative effects in the heart [73,74]. For example, as previously described for ADSCs [96,97], released vesicles induced a strong attenuation of inflammation in infarcted rats, reflected in the reduced serum levels of IL-6, IL-1β, TNF-α, and IFN-γ associated with anti-inflammatory macrophage polarization. In parallel, vesicles improved cardiac function and reduced collagen fiber accumulation [73]. IV administration of ADSC-EVs was also effective at attenuating cardiac I/R associated infarct expansion and apoptosis, with a concomitant reduction of serum levels of cardiac damage specific markers [74].

Another source of MSCs is the human umbilical cord (hucMSCs) [75,76,77]. In a pioneer study, Sun et al. confirmed the safety of hucMSC-derived EVs (hucMSC-EVs) administration on healthy rabbits and infarcted rats [75]. Going a step further, hucMSC-EVs influence fibroblast phenotypic differentiation and function during the inflammatory phase after MI, stimulating the transition from fibroblast to myofibroblast, attenuating inflammation and protecting cardiomyocytes [76]. When the authors investigated the mechanisms involved in these cardioprotective effects, they found that vesicles promoted *Smad7* expression by inhibiting miRNA-125b-5p, which is usually upregulated in acute MI patients [77].

### 3.2. Cardiac Cell-Derived Extracellular Vesicles

#### 3.2.1. Cardiosphere-Derived Extracellular Vesicles

CDCs are multipotent, stromal/progenitor cells derived from heart tissue, with a distinctive antigenic profile (CD105^+^, CD45^−^, CD90^low^), which have shown promising results for myocardial ischemia treatment [98,99,100,101]. The injection of CDCs in murine and porcine MI models ameliorated cardiac dysfunction and reduced scar size, stepping further into phase I/II clinical trials for MI therapeutics [6]. Recently, the effects of transplanted cells were shown to be recapitulated by the administration of their secretome, including EVs [7,102]. According to this study, CDC-derived EVs (CDC-EVs) mimic the cardioprotective effects of their parent cells, as they reduced infarct size 48h after reperfusion in rats subjected to 45 min of coronary artery occlusion. This outcome was observed using cells of rat or human origin, showing that the effect of CDC-EVs is inherent to their cellular origin [81]. Moreover, pre-treatment of CDCs with an exosome formation inhibitor reversed the cardioprotective effects associated with CDCs, confirming the pivotal role of secreted EVs.

The beneficial effects of CDC-derived EVs have been also observed in larger animals, where an increase in the ejection fraction and attenuation in microvascular occlusion and infarct size were reported by different groups [79,80,81]. Along the same lines, a reduction in collagen deposition has been described in the infarct, border, and even in the remote zones in the chronic phase of the MI [79], in a way that is similar to observations in murine models.

CDC-EVs’ regenerative effects are mainly attributed to how they influence inflammatory processes in the receiving cells. As a proof-of-concept, treatment of cardiac macrophages isolated from infarcted rats with CDC-EVs induced a reduction in pro-inflammatory gene expression, such as *Nos2* and *Tnf* [81]. Similarly, murine bone-marrow-derived cells primed with CDC-EVs increased their expression of anti-inflammatory genes, such as *Arg1*, *Ilr4*, *Tgfb1*, and *Vegfa*. A similar trend was observed in a porcine MI model [81]. Accordingly, transcriptomic studies suggest that non-primed macrophages are maintained in a resting state, while CDC-EV-treated macrophages resemble polarized states. Moreover, infarcted rats treated with CDC-EVs showed a reduction in CD68^+^ macrophages in the infarct border zone 48 h after MI [81]. In another study, anti-inflammatory M2 monocytes were increased in peripheral blood in CDC-EV-treated pigs, while arginase-1 was increased in pericardial fluid [80].

A growing interest in the cargo of EVs has especially led to studies of miRNA content and its function. In CDC-EVs, miR-181b was shown to be a mediator of macrophage polarization in vitro and to be involved in a cardioprotective effect in vivo [81]. Another miRNA enriched in CDC-EVs is miR-126 [81], which has been shown to protect myocardial cells from apoptosis, inflammation, fibrosis, and impaired angiogenesis [103]. Another type of RNA that was also found to be enriched in CDC-EVs is Y RNA. These are non-coding RNAs implicated in DNA replication and RNA quality control. Fragments of Y RNAs have been identified as abundant components in the blood and tissues of several mammals, and have been suggested to have a potential diagnostic value [104]. Full-length Y RNA stability is associated with their encapsulation in EVs [105]. A Y RNA fragment, named EV-YF1, was found to be the most abundant in CDC-EVs. Interestingly, the difference in potency of CDC lines after intramyocardial delivery was associated with variation in EV-YF1 abundance in CDC-EVs [106]. In this sense, the injection of EV-YF1 into the left ventricular cavity of I/R-injured rats reduced myocardial damage and induced the expression of *Il10*, compared with control animals [106].

#### 3.2.2. Cardiac Progenitor Cell-Derived EVs

Cardiac progenitor cells (CPCs), considered to be multipotent stem cells [107], are included in cardiac cell therapy studies due to their capacity to differentiate in vitro into beating cardiomyocyte-like cells [108] and vascular cells when transplanted into infarcted hearts [109]. CPC-derived EVs’ (CPC-EVs’) reparative capacity was confirmed after assessing their effectiveness in infarcted mice, where they showed a similar outcome to CPC-treated mice. Moreover, CPC-EVs were shown to carry endoglin, which activated endothelial cells to promote angiogenesis [83]. Accordingly, the cargo of CPC-EVs was found to be enriched in other molecules such as antiapoptotic and proangiogenic miRNAs, e.g., miR-146a-3p, miR-132, and miR-210 [102]. When comparing EVs isolated from CPCs cultured in normoxia and hypoxia, the latter exerted a greater angiogenic effect in vitro than its counterpart, mediated in part by an increase in miR-210 [84]. In the same study, authors showed that the expression of fibrotic genes in the infarcted heart of rats treated with hypoxic CPC-EVs was reduced in comparison with normoxic CPC-EV treatment, corroborating the effect of cell conditioning on the cargo of secreted EVs [84].

Evidence showed differences in EVs’ cardioprotective effect depending on their parental cells. As an example, infarcted rats treated with CPC-EVs presented a reduction in scar size and a decrease in CD68^+^ macrophages in comparison with rats treated with BM-MSC-EVs [85]. This beneficial effect was associated with the presence of pregnancy-associated plasma protein-A (PAPP-A) in CPC-EVs and its role as a facilitator of the release of IGF-1, a known protein with beneficial effects in the heart [85].

### 3.3. Embryonic and Induced Pluripotent Stem Cell-Derived EVs

#### 3.3.1. Induced Pluripotent Stem Cell-Derived EVs

iPS were established in 2006 after murine fibroblast with *Oct4*, *Sox2*, *Klf4* and c-*Myc* were reprogrammed into a pluripotent state [110]. Their appeal is rooted in the possibility of giving rise to cardiac contractile cells and replacing the compromised cardiomyocyte pool with a suitable subtype of cardiomyocytes [111,112]. For this reason, iPS have been used for cardiac cell-therapy studies in the last few years [113]. However, iPS can introduce additional complications due to their immature developmental stage, risk of tumorigenesis and immune rejection, and defects in cardiac electrophysiology, possibly resulting in arrhythmias in the recipient patient [8,114]. Nonetheless, phase 1 clinical trials using cell-sheet iPS for severe ischemic cardiomyopathy are currently ongoing (NCT03763136).

Similar to other cell types, iPS-derived EVs (iPS-EVs) present analogous beneficial effects to their parent cells. The first study correlating the safety and efficacy of iPS and iPS-EVs was conducted by Adamiak et al. [88]. Here, an extensive transcriptomic and proteomic study was accomplished on iPS derived from murine fibroblasts and the respective EVs. Both iPS and iPS-EVs were enriched in miRNAs related to angiogenesis, adaptation to hypoxic stress, cell cycle regulation, and aging processes. Remarkably, certain miRNAs related to cell proliferation, differentiation, apoptosis, and maintenance of self-renewal and pluripotency, such as let-7, miR-145, miR-17-92 cluster, or miR-302a-5p, were exclusively detected in iPS-EVs. Additionally, proteomic studies revealed that some proteins involved in wound healing and cell differentiation were enriched in iPS-EVs. When translated to in vivo studies, iPS and iPS-EVs ameliorated cardiac function, increased systolic function and infarct wall thickness, induced angiogenesis, and reduced apoptosis in a mouse MI model, with superior outcomes in iPS-EV-treated mice [88]. Similar outcomes were reported by Harane and others [86]. After injecting EVs from iPS-derived CPC in the peri-infarct area of immunocompromised mice, left ventricular function was better preserved and ejection fraction improved when compared to iPS-CPC or iPS-derived cardiomyocytes (iPS-CM) treated mice. Regarding the iPS-CPC-EV miRNA content, EVs were enriched in miRNAs related to cell growth, proliferation, survival, metabolism, angiogenesis, and vasculogenesis (e.g., miR-92a, miR-24-3p, miR-93-5p, miR-20b-5p, miR-107, miR-26a-5p, miR-16-5p and miR-130b-3p) when compared to their parental cells. Furthermore, iPS-CM did not achieve recovery in injured hearts, possibly due to the lack of paracrine secretion observed in vitro [86].

Recently, Harane et al. found that no humoral or cellular immune response was detected in iPS-CPC-EVs or parental cell-treated mice in chronic and acute MI models [87]. Thus, iPS-CPC-EVs induced immune-related signaling pathways, triggering tissue repair in the injured heart. In the acute model characterized by a strong inflammatory reaction, neutrophil numbers were decreased as well as the expression of pro-inflammatory cytokines, such as IL-1α, IL-2 and IL-6, while anti-inflammatory IL-10 increased. In the chronic model, pro-inflammatory monocytes and cytokines, namely IL-1α, IL-1β, TNFα and IFNγ, were decreased in the infarcted area. When studied in vitro on peripheral blood mononuclear cells, neither EVs nor parent cells induced T-cell proliferation, but iPS-CPC induced T-cell immune response and activated natural killer (NK) cells [87].

#### 3.3.2. Embryonic Stem Cell-Derived EVs

Other promising pluripotent stem cells that have been extensively used in cardiac cell therapy are ESCs. ESCs derive from the inner cell mass of the developing embryo. ESCs have the potential to maintain a prolonged undifferentiated state and can be used as raw material to obtain distinct cell types [115]. Several studies describe cardiac improvements after treating infarcted animals with ESCs, although these cells were vulnerable to the inflammatory and scarred environment in which they are implanted.

EVs derived from ESC (ESC-EVs) showed advantageous cardiac performance in MI animal models. Left ventricular contractility and ejection fraction were improved, while left ventricular end-systolic diameter decreased in murine models of MI treated with ESC-EVs [91]. Besides, angiogenic and cytoprotective responses were enhanced in EV-treated animals coupled with induction of resident c-kit^+^ CPC survival and proliferation. Here, miR-294 was proposed to be central in regulating the CPC cell cycle, proliferation, and survival [91].

Likewise, cardiovascular progenitor cells derived from human ESC (hCVPD) improved cardiac function when injected during the subacute phase of I/R in rats [116]. EVs derived from hCVPD (hCVPD-EVs) induced similar outcomes improving cardiac function, reducing fibrotic scar and preserving cardiomyocyte size. Overall, EVs showed a tendency to outperform hCVPD treatment, especially in angiogenic capacity [90]. Recently, normoxic and hypoxic hCVPD-EVs improved cardiac function and reduced scar size 28 days after injection in a MI murine model [89]. Additionally, hCVPD-EVs protected cardiomyocytes from ischemic death, with superior results when obtained from hypoxic cells. The beneficial action of hCVPD-EVs was suggested to come, in part, from a lncRNA, MALAT1, found to be more abundant in hypoxic than in normoxic hCVPD-EVs. MALAT1 may have a cardioprotective role in the ischemic heart by targeting miR-497 and increasing the angiogenic capacity of the injured tissue [89].

## 4. The Use of Extracellular Vesicles as a New Class of Drug Delivery System

### 4.1. Advantages of Using Extracellular Vesicles as Delivery Vehicles

Spurred on by the inherent limitations associated with the delivery of fragile therapeutic molecules to the ischemic myocardium, an enthusiastic search for adequate delivery vehicles has been pursued during the last few years in the cardiac regenerative field. From polymeric microparticles to lipid nanoparticles, including others as hydrogels, several DDS have been explored to effectively deliver a cocktail of proteins to the damaged myocardium, with specific kinetics to trigger a reparative response [117,118,119]. Each of these has been characterized in terms of its advantages and shortcomings, but scientists have failed to reach a consensus on the optimal vehicle for this purpose. Moreover, the clinical translation of traditional DDS has been limited and to date, only a few have met with partial success [120,121]. The latest trends, driven by the recent discovery of the key role of EVs in naturally-occurring intercellular communication, have postulated EVs as a potentially advantageous new class of DDS, which are promising to bring significant improvements to the field.

One of the main concerns related to the delivery of therapeutic molecules encapsulated into current DDS is the difficulty of overcoming biological barriers for targeted and intracellular delivery of the cargo, which is needed in most cases to induce a cellular reparative response. By contrast, the exceptional properties of EVs for drug delivery compared with currently available DDS are mainly associated with their biological origin. Firstly, EVs possess the ability to cross biological barriers, have intrinsic cell targeting properties, and transfer their biological cargo to recipient cells, emerging as efficient vehicles [122,123]. Secondly, EVs protect loaded molecules from degradation and are stable in biological fluids, which makes them capable of traveling long distances in blood circulation to reach distal cells and tissues [124]. Finally, the administration of EVs triggers a low or no immunogenic reaction (depending on the nature of the EVs and the cell source, and especially when used autologously) and they are non-toxic due to their natural origin [122]. These advantages pave the way for the use of EVs as next-generation DDS for cardiac applications. Of special importance, their natural cargo could play a double-edged sword role in this context. Although EVs contain a wide variety of molecules with therapeutic potential, this cargo is heterogeneous and depends on the donor cell, an aspect that could hamper their application as DDS.

### 4.2. Bioengineered EVs for Cardiac Delivery

Beyond EVs’ intrinsic natural properties that make them excellent candidates for molecule delivery, emerging approaches focus on the bioengineering of native EVs to improve the above-mentioned inherent properties. Innovative technologies are directed towards the tuning of the EVs’ cargo and surface to develop naturally-inspired vehicles that efficiently deliver their biological cargo to the targeted cells. Specifically, EVs can be modified by the loading of therapeutic molecules as well as by the incorporation of surface molecules to increase their in vivo stability, their bioactivity, and to improve their biodistribution and on-target presentation to specific tissues and cells [5]. Studies from the last five years using bioengineered EVs for cardiac repair are summarized in this section and in Table 3.

#### 4.2.1. Cargo Loading

To date, the therapeutic potential of bioengineered EVs has been poorly addressed in the cardiac regenerative field due to the lack of efficient tools for modifying EVs. However, some studies have proven in the last few years that customization of EVs is a feasible strategy to harness these vesicles as natural DDS. In this context, the most commonly explored approach has been the loading of therapeutics into EVs, which act as natural delivery vehicles (Figure 3). The main challenge surrounding this strategy is the efficient passing of the drugs of interest across the EV membrane. To achieve this, EV loading may be performed by two different approaches. One approach is based on the incorporation of the molecules of interest into the donor cells, which, using the endogenous mechanisms, may result in EV loading and secretion with that specific therapeutic molecule. For example, several therapeutic miRNAs have been loaded into EVs by this method and administered to animal models for cardiac rescue. Song et al. transfected a human embryonic kidney cell line (HEK293T) with anti-apoptotic miRNA-21 and successfully obtained miRNA-21-containing EVs [126]. When compared to liposomes and polymeric polyethylenimine carriers, EVs protected miRNA-21 from degradation more efficiently, preserving it unchanged. Importantly, miRNA-21 was efficiently transferred to cardiomyocytes and endothelial cells after local administration of EVs in the ischemic myocardium of mice, which inhibited cell apoptosis and increased cardiac function [126]. Similarly, inhibition of cardiomyocyte apoptosis and improvement of cardiac function was achieved in another study, where MSC-derived EVs were loaded with miRNA-338 by prior cell transfection using Lipofectamine 2000, followed by local injection of EVs in the heart [127]. MSC lipofection was also used by Li et al. to obtain miRNA-301-loaded EVs for cardiac delivery, resulting in an improved cardiac function and reduced infarct size and myocardial autophagy in rats with MI [128]. Other approaches leveraged on lentivirus for MSCs transduction with miRNA-181a and subsequent isolation of miRNA-181a-loaded EVs, which were capable of inhibiting the inflammatory response in a MI mouse model [129].

Besides intramyocardial administration, EVs have also been explored as therapeutic miRNA delivery vehicles for cardiac repair by IV injection. In one approach, Wang et al. loaded miRNA-101a, a key inhibitor of fibrosis, into MSC-derived EVs by previous cell electroporation, to increase EVs therapeutic potential. In this case, the authors focused their attention not only on achieving heart repair but also on developing a non-invasive strategy. Despite only 4% of injected EVs reached the ischemic myocardium, they remained therapeutically active as they were responsible for decreasing infarct size, fibrosis, and improving heart function while inducing an anti-inflammatory effect [130]. In another non-invasive strategy, Luo et al. improved cardiac repair in rats after tail-vein injection of ADSC-derived EVs loaded with miRNA-126 compared to non-transfected ADSC-EVs. In this case, loading was also performed before EV isolation by ADSCs lipofection [103]. Altogether, these studies show the potential of EVs as a tool for the cardiac delivery of miRNA and other molecules.

The second approach is based on EV loading after isolation. Here, loading mechanisms may be passive, more suitable for hydrophobic drugs, where simple EV co-incubation with the therapeutic molecules results in their diffusion across the lipid bilayer and their encapsulation into the EVs matrix [122]. Alternatively, EV loading with hydrophobic and hydrophilic drugs may be performed by active mechanisms such as electroporation, sonication, or heat shock, which are intended to increase membrane permeabilization and facilitate drug loading. These loading mechanisms have been investigated in other medical fields such as brain delivery [144,145] and cancer [146], but to our knowledge, they still remain unexplored for heart delivery after MI.

#### 4.2.2. Improved Targeting to the Cardiac Tissue

Another EVs modification technology applied in the cardiovascular field is the cell-specific and tissue-specific targeting of EVs to the heart. For that, specific molecules that selectively recognize cardiac epitopes are incorporated on the EVs’ membrane surface. Previous studies have described how EV administration mainly results in vesicle accumulation in the liver and spleen, while only a few vesicles reach the cardiac tissue, which compromises treatment efficacy [130]. Although still in its infancy, the selective EV accumulation in the heart would entail important advances for potential clinical translation. First, it would involve the reduction in the required EV dose, a critical factor since the EV dose is usually limited by EVs’ isolation efficiency. Secondly, it would work in favor of a non-invasive systemic EV administration, a concern of special relevance since local administration in the heart is not usually feasible. Finally, this approach favors EV retention in the heart, which reduces undesirable off-target EV accumulation in peripheral organs and tissues.

As described for the cargo loading, EV targeting may be performed before or after isolation (Figure 4). As an example of modification before isolation, Ciullo et al. transfected CPCs to overexpress CXCR4, which binds SDF-1, a protein that acts as a potent chemoattractant for progenitor cells to the infarcted area. Cell transfection resulted in the CXCR4 enrichment of secreted EVs, which induced greater cardiac recovery and reduced scar size compared to unmodified EVs after IV administration in infarcted rats. In vivo and ex vivo experiments attributed the superior effects described to enhanced EV homing to the heart and uptake by cardiomyocytes [138]. The stimulation of cardiomyocyte proliferation was the targeted purpose of a study led by Wang et al., in which hsa-miR-590-3p, involved in the downregulation of genes that inhibit cell proliferation, was delivered intravenously to a MI animal model using MSC-derived EVs. To stimulate the cardiac-specific location of EVs, donor cells were previously electroporated to express a cTnI-targeting short peptide, which was efficiently incorporated on the isolated EVs membrane [139]. Other studies explored the use of different cardiac-homing peptides such as an ischemic myocardium-targeting peptide [140,147] or cardiac-targeting peptide [141]. Alternatively, a more recent approach used magnetic drug targeting for the accumulation of endogenous EVs in the ischemic myocardium [148].

By contrast, Zhang et al. developed an innovative approach, where EV engineering occurred after isolation [142]. In this study, the authors modified the MSC-EVs membrane by its fusion with monocyte membrane fragments using an incubation–extrusion method. After IV injection in a MI model, the resulting EVs exhibited a higher targeting to the heart than unmodified EVs, mimicking monocyte recruitment to the heart after ischemia. Consequently, the proposed strategy induced better cardiac protection, endothelial maturation during angiogenesis, and modulated macrophage subpopulations [142]. In another attempt to maximize EVs homing to the heart after systemic administration, Vandergriff et al. conjugated cardiac stem cell-derived EVs with the cardiac homing peptide (CHP) through a dioleoylphosphatidylethanolamine *N*-hydroxysuccinimide (DOPE-NHS) linker. Although the exact mechanism behind the interaction between the CHP and the myocardium is unknown, after conjugation to this peptide EVs were retained in the heart more efficiently, which increased cardiac function, reduced fibrosis, and supported cell survival. However, a detailed analysis of mechanisms was lacking in the study [143].

Overall, it should be noted that despite the encouraging advances in the field the wide application of EVs as delivery vehicles and its clinical translation are still hampered and will be determined by the development of efficient and innovative techniques for EVs isolation and modification [149].

## 5. Challenges and Future Directions of Extracellular Vesicle-Based Therapies for Cardiac Repair

The use of EVs for the treatment of MI offers many advantages such as stability in body fluids, site-specific targeting, and ability to deliver therapeutic cargos as previously discussed in this article. However, there are still some challenges, which need to be considered for the translation of EVs into clinical therapies for cardiac repair. These issues have been thoroughly discussed in a European Society of Cardiology Working Group Position Paper together with some problem-solving recommendations [12]. In this section, we highlight the key aspects that must be defined and established when working with EV-based therapies to facilitate their translation to the clinic [150,151,152,153].

Virtually all cell types can secrete EVs into their extracellular environments. However, which cell source is the most appropriate for heart repair applications remains unclear. Careful attention should be paid to the donor cell conditions and status since these have a decisive impact on EVs content and therefore on their therapeutic potential. For example, specifications such as oxygen concentration, serum starvation, cell passage, or confluence should be properly analyzed and provided. This would make possible adequate data interpretation and comparison of the different studies, ultimately facilitating a consensus on protocols for clinical translation. Moreover, the proper characterization of EV-based therapeutics is an important area for work. For instance, data regarding the host-to-donor relationship should be provided. On the other hand, although substantial progress has been made in the isolation and storage procedures, and in the molecular, physical and biological characterization of EVs, more work is still necessary to successfully advance with this therapy.

Large-scale production of EVs is also key. To obtain enough EVs for clinical treatment in humans, a huge quantity of cells is required. Existing EV production methods do not meet the standards for clinical translation. It is, therefore, necessary to develop a production method that assures a high quantity of EVs before reaching the clinic. Currently, some strategies are being investigated to optimize EV production and thus improve the problem of insufficient EV purification [150]. For instance, the use of physical and mechanical stress has increased the production of EVs more than 100-fold [154,155,156], while the use of hollow-fiber culture systems has allowed the continuous production of EVs, increasing its production more than 40-fold [157]. In addition, protocols for purifying EVs should be improved to remove co-eluted particles or soluble factors with biological effects.

As previously described in this review, EVs can be loaded with a range of molecules. The engineered modification methods facilitate the loading of therapeutic cargo molecules including small molecules, proteins, and oligonucleotides into the interior or surface of the EVs. Protocols that maximize the loading efficiency without compromising the cargo stability or the EVs are needed in this respect. At the same time, we must not forget that EVs carry inherently bioactive molecules such as proteins, nucleotides, or lipids that can be transmitted to recipient cells, causing unwanted side effects and impacting their therapeutic efficacy. In this regard, research is underway to selectively remove undesirable substances from EVs via hypotonic lysis to reduce the unwanted biological effects of the inherent cargos [151,158,159,160].

Evaluation of administration routes, safety, toxicity, immunogenicity, and pharmacokinetics of EV-based therapeutics in clinically relevant animal models of MI, such as large animals with comorbidities, is mandatory to support the clinical translation of these therapies. Of note, the administration route might greatly influence the outcome when EVs are used. In animal models of MI, EVs are generally administered locally in the infarcted myocardium or intravenously injected. For patients, direct intramyocardial administration would be generally associated with fewer side effects, increased retention rates at the specific target and reduced vesicle clearance compared to systemic administration. This local administration would be preferable for patients that require a surgical procedure. If not, the IV route could be attractive due to its lack of invasiveness, although it is less effective. Additionally, efforts should be made to achieve in vivo tracking of EVs for an accurate evaluation of their biodistribution and cardiac targeting. A summary of direct labeling of EVs with lipophilic dyes, radionucleotides, or magnetic particles can be found in [161]. Data regarding EV dosage, administration route, timing of delivery, and frequency should be obtained from studies in representative animal models before starting clinical trials evaluating EV-based therapies.

To extend and boost the therapeutic potential of EVs, it is necessary to work on innovative strategies that increase their retention within the infarcted myocardium and limit their systemic biodistribution. A possibility is the combination of EVs with DDS, such as hydrogels [162,163], as they represent a promising tool for cardiac delivery [118]. These novel approaches that rely on biomaterials can be locally administered in the myocardium to provide greater localization and controlled EV release in the target tissue. Thus, the side effects that result from EVs’ biodistribution to peripheral organs is prevented. DDS have been used for decades to ensure improve the therapeutic efficacy and safety of drugs at their site of action [164,165,166,167]. Over recent years, several groups, including ours, have been working on this approach using a variety of biomaterials combined with EVs, with very promising results [168,169,170,171].

Finally, EV-based therapeutics are recognized as biological medicines. They belong to the pharmaceutical class of biologics. Another area to work on is the definition of the active substance, the non-active components and the mode of action, regardless of whether we are working with unmodified EVs or with bioengineered EVs [153]. This is essential information that will be required for their pharmacological classification. Moreover, this point is also important to control the quality of EV-based therapies. In this regard, not only the molecular and physical characterization of EVs is essential, but so is a detailed biological characterization that includes their complexity and heterogeneity, mode of action, potency and quality, among other aspects. This is of primary importance because clinical trials will only start if a mode of action for EV-based therapeutics is hypothesized.

## 6. Conclusions

EVs are becoming a promising approach in cardiac regenerative medicine. As this review has shown, the results in preclinical settings of MI indicate that EVs can modulate inflammation, fibrosis, and apoptosis, and stimulate revascularization. Among these, EVs from MSCs are at present the most widely investigated type used for cardiac repair. However, the delivery of EVs from cardiac-specific cells could bring superior efficacy, by better matching heart requirements. Interestingly, EVs also constitute a new class of drug delivery system due to their intrinsic properties. The surface and cargo of EVs can be adjusted for a greater therapeutic benefit in cardiac repair. Although translational issues such as cell source selection, proper characterization, and large-scale production, among others, need to be solved, the use of EVs for MI is a realistic perspective. The cooperation between researchers, clinicians, and competent authorities is essential to accomplish this.

## Figures and Tables

**Figure 1 nanomaterials-11-00570-f001:**
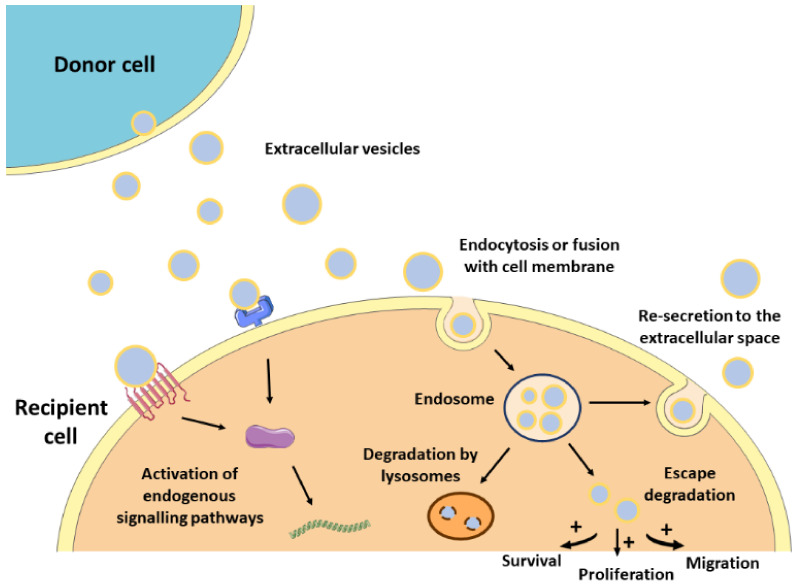
Mechanism of action of EVs. After released from donor cells, EVs may induce a response in recipient cells by different mechanisms. First, EVs may remain at the binding site on the cell membrane eliciting functional responses by activating downstream molecular pathways. Alternatively, EVs may be internalized by endocytosis or fusion with the cell membrane undergoing different intracellular fates. They can be targeted for degradation by lysosomes, they can escape degradation and modulate cell behavior, or they can be re-secreted to the extracellular space.

**Figure 2 nanomaterials-11-00570-f002:**
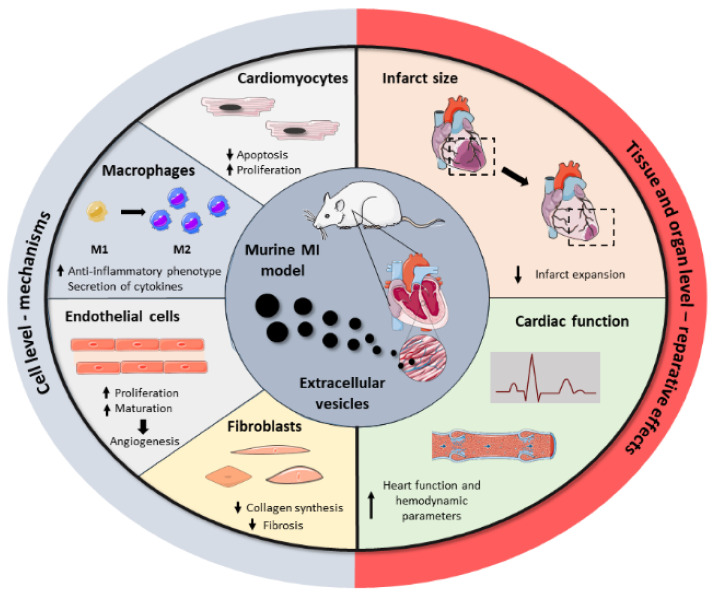
Summary of the beneficial effects of EVs in cardiac repair. Administration of EVs in MI preclinical models showed that EVs modulate a regenerative response in several cardiac cells, including cardiomyocytes, macrophages, endothelial cells, and fibroblasts. Together, these cell-level effects result in the reduction of infarct size and the improvement of cardiac function after MI.

**Figure 3 nanomaterials-11-00570-f003:**
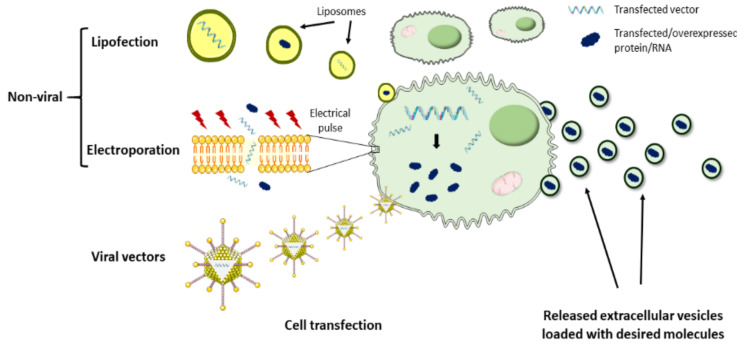
Summary of EVs’ cargo-loading techniques. Several approaches have been explored to efficiently load EVs with therapeutic proteins or RNAs. EVs loading may be performed by modification of donor cells or by engineering isolated EVs. Regarding EVs applied for cardiac repair, recent studies focused on the transfection of parental cells with the desired vectors/molecules by lipofection and electroporation, or in donor cell transduction using viral vectors. Subsequently, the transfected/transduced parental cells use their endogenous machinery to produce EVs loaded with that specific molecule.

**Figure 4 nanomaterials-11-00570-f004:**
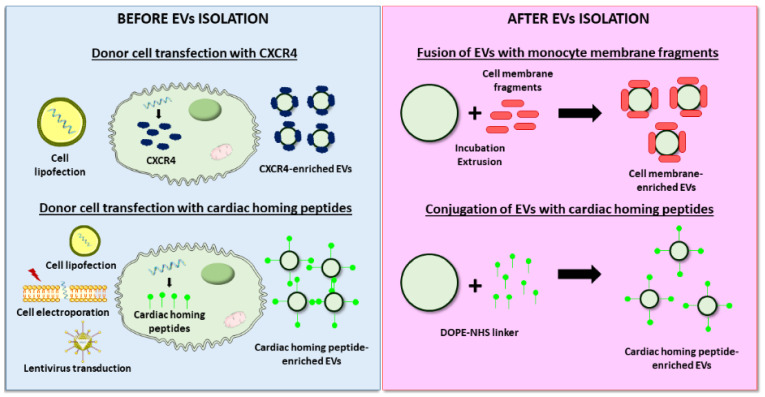
Examples of EVs surface functionalization strategies to improve cardiac targeting. Modification of EV surface can be performed before or after isolation. Regarding the first approach, donor cells are transfected/transduced to overexpress specific molecules that target cardiac epitopes on the released EV’s surface. On the other hand, different techniques can be used to incorporate cardiac-targeting molecules on the EV’s surface after isolation, such as fusion of EVs with monocyte membrane fragments by incubation/extrusion or conjugation of cardiac homing peptides via a dioleoylphosphatidylethanolamine *N*-hydroxysuccinimide (DOPE-NHS) linker.

**Table 1 nanomaterials-11-00570-t001:** Requirements for extracellular vesicles’ separation and classification based on their physical and biochemical properties as well as donor cell conditions.

**Physical properties**	**Size** [10]	Small EVs	<100 nm	
		Small/medium EVs	<200 nm	
		Medium/large EVs	>200 nm	
	**Density** **(in sucrose)** [23]	Low	1.13–1.19 g/mL	
		Medium	1.16–1.28 g/mL	
		High	>1.28 g/mL	
**Biochemical composition**	**Surface antigens** [10]	Tetraspanins MHC class I Integrins Transferrin receptor LAMP1/2 Heparan sulfate	Proteoglycans EMMPRIN ADAM10 GPI-anchored 5ʹnucleotidase CD73 Complement-binding proteins CD55 and CD59 Sonic hedgehog	
	**Lipids** [10,24]	Phosphatidylserine Phosphatidylinositol Phosphatidylethanolamine Phosphatidylcholine	Cholesterol Ceramide Diacylglycerol Glycosphingolipids	
	**Internal cargo** [10,25,26]	Proteins	TSG101 ALIX VPS4A/B ARRDC1 Flotillins-1 and 2	Caveolins Annexins Heat shock proteins HSC70 and HSP84 Syntenin
		Cardiac-related miRNAs	let-7 miR-16 miR-17-92 miR-19b miR-20a/b miR-21a miR-24 miR-26a miR-34 miR-93 miR-94a miR-107a miR-125b miR-126	miR-130a/b miR-132 miR-143 miR-145 miR-146a miR-181b miR-182 miR-208a miR-210 miR-214 miR-294 miR-302a miR-451
**Conditions at EVs harvest**	**Cell culture conditions** [10]	Normoxia Hypoxia Surface coating	Treatment Grade of confluency Passage number	
	**Donor status** [10]	Age Biological sex Circadian variation Body mass index	Pathological/healthy condition Exercise level Diet Medication	

ADAM10: ADAM metallopeptidase domain 10; ALIX: ALG-2-interacting protein X; ARRDC1: arrestin domain-containing protein 1; EMMPRIN: extracellular matrix metalloproteinase inducer; EVs: extracellular vesicles; GPI: glycosylphosphatidylinositol; LAMP: lysosomal-associated membrane protein; MHC: major histocompatibility complex; miRNA: microRNA; TSG101: tumor susceptibility gene 101; VPS4A/B: Vacuolar Protein Sorting 4 Homolog A/B.

**Table 2 nanomaterials-11-00570-t002:** Representative preclinical efficacy studies from the last five years, using extracellular vesicles as therapeutic agents for myocardial infarction.

Cell Source	Isolation Method	Animal Model	Dose	Administration Route and Time Post-MI	Reparative Effect	Molecule/Mechanism Involved	Ref
**MSCs**							
**Rat BM-MSCs**	Total Exosome Isolation Kit (Invitrogen)	Rat, permanent	20 µg	IM; immediate	Improved cardiac function Reduced fibrosis Reduced inflammation	-	[62]
**Mouse BM-MSCs**	Density- gradient UC	Mouse, I/R	50 µg	IM; immediate after reperfusion	Reduced infarct size Alleviated inflammation (polarization of macrophages to M2 phenotype)	Inhibition of TLR4 by miR-182	[63]
**Proinflammatory rat BM-MSCs**	Density-gradient UC	Mouse, permanent	50 µg	IM; immediate	Reduced inflammation Anti-inflammatory macrophage polarization Reduced cardiomyocyte apoptosis	Suppression of NF-κB and regulation of AKT1/AKT2	[64]
**BM-MSCs**	UC	Rat, permanent	10 µg EVs (and 2×10^6^ BM-MSCs)	IM; at 30 min	Improved cardiac function Reduced infarct size and fibrosis Increased vascularization Reduced inflammation Enhanced recruitment of IV-infused MSCs	-	[65]
**ATV-pre-treated rat BM-MSCs**	UC	Rat, permanent	10 µg	IM; immediate	Improved cardiac function Reduced infarct size Decreased cardiomyocyte apoptosis Increased angiogenesis	lncRNA H19 and miR-675	[66]
**Mouse BM-MSCs**	UC	Mouse, permanent	-	IV; immediate and day 6	Improved cardiac function Angiogenesis Reduced fibrosis	miR-210 and Efna3 gene suppression	[67]
**Mouse BM-MSCs**	UC	Mouse, permanent	EVs derived from 2×10^7^ cells	IM; immediate	Improved cardiac function Angiogenesis Decreased scar size Reduced cardiomyocyte survival Activation of resident CPCs	miR-210	[68]
**Rat BM-MSCs**	Total Exosome Isolation Kit (Invitrogen)	Rat, I/R	5 µg	IM; prior to reperfusion	Decreased cardiomyocyte apoptosis Reduced infarct size Improved heart function by an enhanced autophagy	AMPK and AKT pathways	[69]
**Mouse BM-MSCs**	UC	Mouse, I/R	12.5 µg/ 5.62×10^5^ EVs	IM; 24h prior to ischemia	• Decreased infarct size	Reduced expression of pro-apoptotic genes PDCD4, PTEN, Peli1 and FasL via miR-21a-5p	[70]
**Mouse BM-MSCs**	UC	Mouse, permanent	200 µg	IM; immediate	Improved cardiac function Reduced infarct size	miR-125b	[71]
**BM-MSCs**	ExoQuick	Rat, permanent	-	IM; immediate	Reduced infarct size Alleviated cardiomyocyte apoptosis Improved cardiac function	miR-24	[72]
**Rat ADSCs**	UC	Rat, permanent	2.5×10^12^ particles	IV; at 1h	Decreased fibrosis Decreased cell apoptosis Attenuated inflammation via anti-inflammatory macrophage polarization Improved cardiac function	S1P/SK1/S1PR1 activation	[73]
**Rat ADSCs**	Ultrafiltration and UC	Rat, I/R	400 µg	IV; at reperfusion	Reduced infarct area Attenuated apoptosis Reduced serum levels of cardiac damage markers	Wnt/β-catenin activation	[74]
**Human umbilical cord MSCs**	Density-gradient UC	Rat, permanent	400 µg and 800 µg	IV; once daily for 7 days	• Safety: no effect on hemolysis, no vascular and muscle stimulation, no side effects on hematology indexes, liver and renal function, and protective effect on weight loss	-	[75]
**Human umbilical cord MSCs**	ExoQuick-TC (System Biosciences)	Rat, permanent	400 µg	IM; immediate	Increased density of myofibroblasts Attenuated inflammation Reduced cardiomyocyte apoptosis	-	[76]
**Human umbilical cord MSCs**	Density-gradient UC	Rat, permanent	400 µg	IV; immediate	Improved cardiac function Increased cardiomyocyte survival	Upregulation of Smad7 by inhibition of miR-125b-5p	[77]
**Cardiac MSCs**	Precipitacion with PEG	Mouse, permanent	50 µg	IM; immediate	Improved cardiac function Increased scar thickness Angiogenesis Cardiomyocyte proliferation	-	[78]
**CDCs**							
**Human CDCs**	Ultrafiltration and precipitation with PEG	Pig, I/R	7.5 mg	IC; 30 min after reperfusion IM; 30 min after reperfusion	Decreased infarct size and preserved LV function Reduced leukocyte infiltration Reduced fibrotic mass Higher arteriolar density	-	[79]
**Porcine CDCs**	Ultrafiltration followed by Field-Flow Fractionation	Pig, I/R	9.16 mg	IM; at 72h after reperfusion	Inconclusive; tendency to reduce infarct size and increase cardiac function Increased M2 macrophages	-	[80]
**Human CDCs**	Ultrafiltration and PEG precipitation	Pig, I/R	7.5 mg	IM; at 20 min after reperfusion	Preserved cardiac function Reduced microvascular occlusion Attenuated infarct size Reduced CD68^+^ macrophages infiltration	Regulation of gene expression by miRNA	[81]
**Human CDCs**	Ultrafiltration and precipitation with PEG	Rat, I/R	350 µg	IM; at 30 min after reperfusion	Preserved cardiac function Reduced infarcted area	-	[81]
**Human CDCs**	ExoQuick (precipitation)	Rat, permanent	250 µg	IM; at 4 weeks	Improved cardiac function Reduced scar mass Increased wall thickness Increased capillary and microvessel density	Regulation of gene expression by miRNA	[82]
**CPCs**							
**Human CPCs**	Density-gradient UC	Mice, permanent	8 µg	IM; at 15 min	Reduced infarct size Increased proliferation of cardiomyocytes and endothelial cells	Activation of endoglin in endothelial cells	[83]
**Rat CPCs**	UC	Rat, I/R	5 µg/kg	IM; during reperfusion	Reduced infarct size Increased cardiac contractility	Decreased levels of collagen I, collagen III, vimentin and CTGF Regulation of gene expression via miRNA	[84]
**Human CPCs**	UC	Rat, permanent and I/R	10^11^ particles	IM; at 1h after permanent ligation or at reperfusion	Increased cardiac function Reduced scar size Increased blood vessel density Decreased CD68^+^ macrophages	miR-146a-3p, miR-132, and miR-181a PAPP-A IGF-1	[85]
**iPS**							
**Human iPS**	UC	Mouse, permanent	3×10^10^ particles	Transcutaneous echo-guided IM; at 3 weeks	• Increased cardiac function	Regulation of gene expression via miRNA	[86]
**Human iPS**	UC	Mouse, permanent	100 µg (10^10^ particles)	IM; at 2 days or 3 weeks	No detectable humoral or immune response Decreased pro-inflammatory monocytes and cytokines	-	[87]
**Mouse iPS**	UC	Mouse, I/R	100 µg	IM; at 48h after reperfusion	Preserved cardiac function Improved systolic infarct wall thickness Smaller LV end-systolic volume Reduced apoptosis in myocytes Increased capillary density No tumor formation	Regulation of gene expression via miRNA and metabolic regulation via protein delivery (in silico analysis)	[88]
**ESC**							
**Human ESC**	UC	Mouse, permanent	20 µg	IM; immediate	Improved cardiac function and LV systolic dimension Reduced scar size Decreased cardiomyocyte apoptosis Higher number of endothelial cells	Targeting miR-497 by lncRNA MALAT1	[89]
**Human ESC**	UC	Mouse, permanent	-	Transcutaneous echo-guided IM; at 2-3 weeks	Decreased LV end-systolic and diastolic volume Reduced fibrosis Smaller cardiomyocytes	Gene regulation of DNA repair, cell survival, cell cycle progression and cardiomyocyte contractility (*in silico*)	[90]
**Mouse ESC**	UC	Mouse, permanent	-	IM; immediate	Enhanced contractility and decreased LV end-systolic diameter Increased capillary density Reduced apoptosis Elevated cardiomyocyte proliferation	Regulation of CPC cell cycle and association with proliferation and survival mediated by miR-294	[91]

ADSCs: adipose tissue-derived mesenchymal stem cells; ATV: atorvastatin; BM-MSCs: bone marrow-derived mesenchymal stromal cells; CDCs: cardiosphere-derived cells; CPCs: cardiac progenitor cells; CTGF: connective tissue growth factor; ESC: embryonic stem cells; EVs: extracellular vesicles; I/R: ischemia/reperfusion; IC: intracoronary; IGF-1: insulin-like growth factor-1; IM: intramyocardially; iPS: induced pluripotent stem cells; IV: intravenously; lncRNA: long non-coding RNA; LV: left ventricle; MALAT1: metastasis-associated lung adenocarcinoma transcript 1; MI: myocardial infarction; miRNA: microRNA; MSCs: mesenchymal stromal cells; NF-κB: nuclear factor kappa B; PAPP-A: pregnancy-associated plasma protein A; PEG: polyethylene glycol; S1P: sphingosine 1-phosphate; S1PR1: sphingosine-1-phosphate receptor 1; SK1: sphingosine kinase 1; Smad7: mothers against decapentaplegic homolog 7; TLR4: toll-like receptor 4; UC: ultracentrifugation.

**Table 3 nanomaterials-11-00570-t003:** Representative preclinical efficacy studies from the last five years using bioengineered extracellular vesicles as a novel class of drug delivery systems for myocardial infarction.

Cell Source	Isolation Method	Modification	Method	Animal Model	Dose	Administration Route and Time Post-MI	Reparative Effect	Mechanism	Ref
**Cargo loading**									
**Cardiac MSCs**	SEC	Notch1 overexpression	Adenovirus	Mouse, permanent	2×10^10^ particles	IM, at 10 min	Decreased apoptosis and fibrosis Induced cardiomyocyte proliferation Increased vascularization Improved cardiac function	-	[125]
**HEK293T**	UC	miR-21 overexpression	-	Mouse, permanent	20 µg	IM, immediate	Inhibited cell apoptosis Increased angiogenesis Reduced scar thickness Improved cardiac function	PDCD4	[126]
**Rat BM-MSCs**	Exosome Isolation Reagent (RiboBio)	miR-338 overexpression	Lipofection	Rat, permanent	-	IM, immediate	Improved cardiac function Inhibited cardiomyocyte apoptosis	MAP3K2/JNK pathway	[127]
**Rat BM-MSCs**	Exosomes isolation kit (Thermo Fisher Scientific)	miR-301 overexpression	Lipofection	Rat, permanent	-	IM, at 30 min	Inhibited myocardial autophagy Decreased infarct area Improved cardiac function	-	[128]
**Human umbilical cord MSCs**	UC	miR-181a overexpression	Lentivirus	Mouse, I/R	200 µg	IM, at reperfusion	Improved cardiac function Reduced infarct area Reduced inflammation	c-Fos inhibition	[129]
**Human BM-MSCs**	Total Isolation Reagent (Thermo Fisher Scientific)	miR-101a overexpression	Electroporation	Mouse, permanent	2 mg/kg	IV, at days 2 and 3	Increased number of anti-inflammatory macrophages Decreased infarct size and fibrosis Improved cardiac function	-	[130]
**Rat ADSCs**	ExoQuick-TC (System Biosciences)	miR-126 overexpression	Lipofection	Rat, permanent	400 µg	IV, immediate	Decreased infarct area and fibrosis Reduced inflammation Increased vasculogenesis	-	[103]
**Human umbilical cord MSCs**	UC	SDF1 overexpression	Lipofection	Mouse, permanent	-	IM, immediate	Decreased infarct size Reduced tissue damage Reduced levels of inflammatory cytokines	-	[131]
**Human umbilical cord MSCs**	UC	TIMP2 overexpression	Lentivirus	Rat, permanent	50 µg/ml	IM, immediate	Improved cardiac function Decreased infarct size and fibrosis Reduced ventricular dilation Reduced cell apoptosis Increased angiogenesis Inhibited cardiac fibroblast proliferation	Akt/Sfrp2 Pathway	[132]
**Human MSCs**	Exosome isolation kit (Invitrogen)	LncRNA KLF3-AS1 overexpression	Lipofection	Rat, permanent	40 µg	IV, at 1 week	Reduced infarct area Ameliorated apoptosis Decreased inflammation	miR-138-5p and Sirt1	[133]
**Mouse BM-MSCs**	ExoQuick TC (SBI)	GATA4 overexpression	Lentivirus	Mouse, permanent	20 µg	IV, at 48h	Increased cardiac function Angiogenesis Increased number of c-kit^+^ cells Decreased apoptotic cardiomyocytes	-	[134]
**Human umbilical cord MSCs**	Density-gradient UC	Akt overexpression	Adenovirus	Rat, permanent	400 µg	IV, immediate	Improved cardiac function Decreased cell apoptosis Enhanced vasculogenesis	PDGF-D activation	[135]
**MSCs**	ExoQuick-TC (System Biosciences)	miR-150-5p overexpression	Lentivirus	Rat, I/R	5.8×10^12^ particles	IM, at 10 min before reperfusion	Improved cardiac function Suppressed myocardial remodeling Reduced cardiomyocyte apoptosis	TXNIP downregulation	[136]
**Rat ADSCs**	ExoQuick (System Biosciences)	miR-146a overexpression	Lipofection	Rat, permanent	400 µg	IV, immediate	Reduced apoptosis Inhibited inflammation response Decreased fibrosis and infarct volume	EGR1 downregulation	[137]
**Targeting**									
**Human CPCs**	UC	CXCR4	Cell lipofection	Rat, I/R	2×10^11^ particles	IV, at 3h after reperfusion	Increased blood vessel density Reduced infarct size Improved cardiac function	ERK1/2 activation	[138]
**Rat BM-MSCs**	UC	cTnI-targeted short peptide	Cell electroporation	Rat, permanent	200 µg	IV, immediate	Cardiomyocyte proliferation Reduced adverse remodeling Restored cardiac function	-	[139]
**Mouse BM-MSCs**	Total Exosome Isolation Reagent (Invitrogen)	Cardiac homing peptide (CSTSMLKAC)	Lentivirus transfection of cells	Mouse, permanent	4×10^9^ particles/50 μg	IV, immediate	Attenuated inflammation Reduced cell apoptosis Increased vasculogenesis Reduced fibrosis Improved cardiac function	-	[140]
**HEK293**	Tangential Flow Filtration	Cardiac targeting peptide (APWHLSSQYSRT)	Cell lipofection	Mouse, no infarct	150 µg	IV, no infarct	• 15% enhanced heart delivery of EVs	-	[141]
**Rat BM-MSCs**	UC	Monocyte membrane	EVs and monocyte membrane fusion	Mouse, I/R	10 µg	IV, at days 2 and 3	Enhanced specific cardiac delivery Angiogenesis Endothelial maturation Anti-inflammatory macrophage modulation Reduced fibrosis Improved cardiac function	-	[142]
**CDCs**	Ultrafiltration	Cardiac homing peptide (CSTSMLKAC)	EVs conjugation via a DOPE-NHS linker	Rat, I/R	6×10^9^ particles	IV, at 24h	Increased cell proliferation Angiogenesis Improved heart function Reduced infarct size and fibrosis	-	[143]

ADSCs: adipose tissue-derived mesenchymal stem cells; BM-MSCs: bone marrow-derived mesenchymal stromal cells; CPCs: cardiac progenitor cells; cTnI: cardiac troponin-I; CXCR4: C-X-C chemokine receptor type 4; DOPE-NHS: dioleoylphosphatidylethanolamine *N*-hydroxysuccinimide; EGR1: early growth response factor 1; EVs: extracellular vesicles; HEK293T: Human embryonic kidney 293 cell line; I/R: ischemia/reperfusion; IM: intramyocardially; IV: intravenously; lncRNA: long non-coding RNA; MI: myocardial infarction; miRNA: microRNA; MSCs: mesenchymal stromal cells; PDGF-D: platelet-derived growth factor D; SDF1: stromal-derived factor 1; SEC: size-exclusion chromatography; Sfrp2: secreted frizzled-related protein 2; TIMP2: tissue matrix metalloproteinase inhibitor 2; TXNIP: thioredoxin-interacting protein; UC: ultracentrifugation.

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
