# Peer review of "Extracellular Vesicle-Based Therapeutics for Heart Repair"

_nanomaterials, 2021, doi:10.3390/nano11030570_

Round 1
Reviewer 1 Report
In the present review, Saludas et al. provided an overview of the most recent discoveries regarding the therapeutic potential of EVs for addressing cardiac damage after MI. The review is well written and quite comprehensive.
Author Response
Reviewer 1: In the present review, Saludas et al. provided an overview of the most recent discoveries regarding the therapeutic potential of EVs for addressing cardiac damage after MI. The review is well written and quite comprehensive.
We thank the reviewer for his/her kind comments.
Reviewer 2 Report
An excellent extensive review of the potential of extracellular vesicles for cardiac repair, reviewing the techniques of isolation, modification and existing areas of innovative application.
Very good illustrations
Author Response
Reviewer 2: An excellent extensive review of the potential of extracellular vesicles for cardiac repair, reviewing the techniques of isolation, modification and existing areas of innovative application.
Very good illustrations
We truly appreciate the comment from the reviewer.
Reviewer 3 Report
Writing a review is always a difficult task either it is an original publication or some "judgement" on it. The topic of the review written by Saludas and co-workers is the extracellular vesicle-based therapeutics for heart repair. No question, it is an important issue and hot topic because of the EVs and because of the high burden of heart diseases on human society, too.
The number of cited papers is more than 170, and due to the start of research in this field only nowadays, the majority of these papers were published in the last 5 years. Compared to 170, the about 25 reviews in this list are not many, but focusing on their publication dates - several between 2018-2020 - the question arises which newest data, publications demanded the writing of another one? Whether did our authors find a different viewpoint which can make their review more relevant in this special, heart-repair topic? One thing cannot be denied, they tried hard and worked a lot. The review contains well organized, very useful tables and clear, easily followable figures. Perhaps even another figure which demonstrates the fates of endocyted EVs could be useful, too (2.3. Mechanism of action).
The Chapter 4 and 5 give the real answer for the title, while the previous chapters, in spite of their many important data, could be reconsidered and shortened, because they are a bit lenghty containing some repetitions and can take the focus away.
It is very important to draw the attention to the unsolved tasks before the possible (and expected) clinical, therapeutic application application of EVs and the Authors do not forgive about it.
Practically every text can be written better - including my comments, too - but for me the best part of the paper is chapter 5.
This is when a reader can feel that the authors are really familiar with the topic with its pros and cons and the readers can rely on their opinion regarding the importance of advantages/disadvantages and possibilities of the question under discussion.
Taken together, a shorter Introduction, more focus on the special issues of the heart, much less details about EV preparations and removal the redundant parts would improve the paper significantly. (While the list of the main techniques/methods is great, giving details cannot help a lot, due to the different sources of EVs, either. )
The Conclusion is OK - but the last sentence of the Chapter 5 seems the most powerful sentence for me, which could be use as closing words, too.
Author Response
Author's Response to Decision Letter for Ms. ID: nanomaterials-1077458
Title: EXTRACELLULAR VESICLE-BASED THERAPEUTICS FOR HEART REPAIR
Reviewer 3: Writing a review is always a difficult task either it is an original publication or some "judgement" on it. The topic of the review written by Saludas and co-workers is the extracellular vesicle-based therapeutics for heart repair. No question, it is an important issue and hot topic because of the EVs and because of the high burden of heart diseases on human society, too.
The number of cited papers is more than 170, and due to the start of research in this field only nowadays, the majority of these papers were published in the last 5 years. Compared to 170, the about 25 reviews in this list are not many, but focusing on their publication dates - several between 2018-2020 - the question arises which newest data, publications demanded the writing of another one? Whether did our authors find a different viewpoint which can make their review more relevant in this special, heart-repair topic? One thing cannot be denied, they tried hard and worked a lot. The review contains well organized, very useful tables and clear, easily followable figures. Perhaps even another figure which demonstrates the fates of endocyted EVs could be useful, too (2.3. Mechanism of action).
The Chapter 4 and 5 give the real answer for the title, while the previous chapters, in spite of their many important data, could be reconsidered and shortened, because they are a bit lenghty containing some repetitions and can take the focus away.
It is very important to draw the attention to the unsolved tasks before the possible (and expected) clinical, therapeutic application of EVs and the Authors do not forgive about it.
Practically every text can be written better - including my comments, too - but for me the best part of the paper is chapter 5.
This is when a reader can feel that the authors are really familiar with the topic with its pros and cons and the readers can rely on their opinion regarding the importance of advantages/disadvantages and possibilities of the question under discussion.
Taken together, a shorter Introduction, more focus on the special issues of the heart, much less details about EV preparations and removal the redundant parts would improve the paper significantly. (While the list of the main techniques/methods is great, giving details cannot help a lot, due to the different sources of EVs, either. )
The Conclusion is OK - but the last sentence of the Chapter 5 seems the most powerful sentence for me, which could be use as closing words, too.
We thank the reviewer for his/her thoughtful comments and suggestions to improve the quality of the manuscript. We believe that the clinical application of extracellular vesicles – for addressing myocardial infarction in this case – is a hot topic that is attracting the interest of many researchers and clinicians worldwide. Therefore, the number of related publications is rapidly increasing. As an example, the number of publications per year about “cardiac repair” and “extracellular vesicles” in Pubmed database has almost quintuplicated in the last five years, the period addressed in our article. Furthermore, as this is a very recent field, many aspects regarding the use of extracellular vesicles for cardiac application remain to be elucidated (e.g. cell source, isolation technique, therapeutic doses, route of administration, etc.), with every contribution resulting meaningful nowadays to advance in the field. Therefore, we considered that providing a review article summarizing and discussing the latest findings could be useful. Although there are a few previous recent review articles on this topic, we modestly believe that we provided a helpful, practical, and quite complete review on the most recent discoveries of extracellular vesicles for heart repair. We covered different areas, focusing on cell sources and preclinical findings, and keeping always in mind the potential clinical translation. In this sense, we agree with the reviewer that sections 4 and 5 contain highly relevant information about the application of extracellular vesicles as a new class of drug delivery systems – perhaps the most innovative application – and finally about a critical and important point as it is the unsolved tasks and key factors for the translation to the clinic. However, our objective was not to only cover those relevant topics, but also to provide a simple and clear overview of the pathological condition after myocardial infarction and biological aspects of extracellular vesicles, which could help the reader to gain some basic insights into the topic and not losing the focus in the last sections. With that in mind and concerning the reviewer’s suggestion, we believe that the Introduction section is already focused on heart issues. The only paragraph that could be out of that specific context is the fifth paragraph about bioengineered extracellular vesicles. However, we consider it important to introduce that topic addressed in section 4 in our review. Similarly, we found section 2 of special importance for the reader to delve into the basis of extracellular vesicles, for an improved understanding of the following sections. Following the reviewer’s suggestion, we have shortened sections 2.2. and 2.4. to provide fewer details about EV preparation. Now, it reads as follows:
- Page 4, lines 14-18: << Exosomes, generated within the endosomal system, are intraluminal vesicles formed by the inward budding of the endosomal membrane during maturation of multivesicular endosomes (MVEs) and are released to the extracellular space by fusion of MVEs with the cell membrane [29]. As for microvesicles, they originate by outward budding of the plasma membrane [30]. >>
- Page 6, lines 1-35: << 2.4. Separation and characterization of extracellular vesicles
EVs can be separated from the cell culture medium and most body fluids (liquid biopsy), blood being the most frequently studied source of EVs. Before EVs separation, some preanalytical parameters should be considered [45,46], such as the use of serum-free media or EVs-depleted serum for EVs separation from conditioned medium [10,47]. Moreover, differences in the physicochemical and biochemical properties of the selected separation methods can impact the enriched EVs subpopulations [48–52] and do not enable an absolute purification of EVs from other contaminants [53].
Ultracentrifugation (UC) is the most commonly used EVs separation and enrichment technique based on particle density, involving multiple centrifugation and ultracentrifugation steps [48]. Speeds of 10,000-20,000 g enable the separation of medium/large vesicles, while small size vesicles are recovered at higher speeds (100,000 g).
Size exclusion techniques include ultrafiltration and chromatography. Ultrafiltration is usually based on cellulose filters defined by molecular mass and size exclusion range [49], while size exclusion chromatography (SEC), separates fractions by elution with PBS and has been proven to be reliable and scalable for various applications [54].
Immune affinity isolation is based on the immunolabeling of proteins on the surface of EVs enabling the separation of specific particle subpopulations from other EVs classes, contaminant protein aggregates or lipoproteins. Usually, specific antibodies are conjugated to magnetic beads and EVs are separated using magnets [55,56].
A range of commercial kits are also available, some based on polymer precipitation- methods [49–51] and others on non-precipitation alternatives, for instance, those selective for phosphatidylserine positive vesicles [47]. Alternative or complementary techniques to classical procedures are also emerging, including microfluidics, asymmetric flow field-flow fractionation, or high-resolution flow cytometry [53].
After separation, EVs purity should be tested by the use of multiple complementary methods: (i) western blotting to analyse EVs markers (e.g. CD63, Alix, etc.) and co-isolated contaminants [10]; (ii) Nanoparticle Tracking Analysis (NTA), which can determine particle size and concentration [57]; (iii) conventional transmission electron microscopy (TEM) and the more strongly recommended cryo-TEM [58] or (iv) nanoflow cytometry, that enables the determination of cell surface antigens, the quantification of EVs subpopulations based on parental cell markers [59,60], and the lipid nature of the studied particles with cell-permeant, non-fluorescent pro-dyes [61]. Also, current advances in EVs-adapted proteomic, lipidomic and genomic technologies will greatly help to delimit the molecular signature of EVs subpopulations under research. >>
Also, we agree with the reviewer that a new figure related to the mechanism of action of extracellular vesicles could help the reader to get into the topic. Therefore, we have included below as well as in the revised version of the manuscript the following new figure:
Figure 1. Mechanism of action of extracellular vesicles. After released from donor cells, EVs may induce a response in recipient cells by different mechanisms. First, EVs may remain at the binding site on the cell membrane eliciting functional responses by activating downstream molecular pathways. Alternatively, EVs may be internalized by endocytosis or fusion with the cell membrane undergoing different intracellular fates. They can be targeted for degradation by lysosomes, they can escape degradation and modulate cell behaviour or they can be re-secreted to the extracellular space.
Finally, the final sentence of Chapter 5 has been moved to the end of the conclusion as closing words. The text now reads as follow:
- Page 24, lines 37-38: << Although translational issues such as cell source selection, proper characterization and large-scale production, among others, need to be solved, the use of EVs for MI is a realistic perspective. The cooperation between researchers, clinicians and competent authorities is essential to accomplish this. >>

Reviewer 4 Report
In the manuscript titled Extracellular vesicle-based therapeutics for heart repair, the authors focus on the most recent discoveries regarding the therapeutic potential of EVs for addressing cardiac damage after myocardial infarction. Moreover, also the use of bioengineered EVs for targeted cardiac delivery are reported highlightening both the possibility to exploit extracellular vesicles as drug delivery systems and the most crucial aspects that should be addressed before a widespread translation to the clinical area.
The topic is very interesting, consistant well designed.
I think that the paper is suitable for the pubblication on Nanomaterials.
Author Response
Reviewer 4: In the manuscript titled Extracellular vesicle-based therapeutics for heart repair, the authors focus on the most recent discoveries regarding the therapeutic potential of EVs for addressing cardiac damage after myocardial infarction. Moreover, also the use of bioengineered EVs for targeted cardiac delivery are reported highlightening both the possibility to exploit extracellular vesicles as drug delivery systems and the most crucial aspects that should be addressed before a widespread translation to the clinical area.
The topic is very interesting, consistent well designed.
I think that the paper is suitable for the publication on Nanomaterials.
We thank the reviewer for the kind comments.